# Factors Influencing Healthcare Experience of Patients with Self-Declared Diabetes: A Cross-Sectional Population-Based Study in the Basque Country

**DOI:** 10.3390/healthcare9050509

**Published:** 2021-04-28

**Authors:** Roberto Nuño-Solínis, Sara Ponce, Maider Urtaran-Laresgoiti, Esther Lázaro, María Errea Rodríguez

**Affiliations:** 1Deusto Business School Health, University of Deusto, 48014 Bilbao, Spain; roberto.nuno@deusto.es (R.N.-S.); maider.urtaran@deusto.es (M.U.-L.); 2International Research Projects Office, University of Deusto, 48007 Bilbao, Spain; sponce@deusto.es; 3Faculty of Health Sciences, Valencia International University, 46002 Valencia, Spain; melazaro@universidadviu.com; 4Freelance Researcher, 31007 Pamplona, Spain

**Keywords:** patient-centered care, diabetes mellitus, chronic conditions

## Abstract

Background: Diabetes affects more than 400 million people around the world. Few published studies incorporate questionnaires that comprehensively cover every aspect of a patient’s experience of healthcare. This study analyzes potential differences in the healthcare experience for patients with diabetes based on their sociodemographic, economic, and health-related characteristics from a comprehensive viewpoint in an integrated delivery system. Methods: We used data from the 2018 Basque Health Survey, which includes a questionnaire for the measurement of the experiences of patients with chronic problems. We present descriptive and regression analyses to explore differences by sociodemographic, economic, and health-related characteristics of patients’ experiences with different healthcare services. Results: Having diabetes plus other comorbidities significantly decreases the quality of the experience with all healthcare services and decreases the global healthcare experience score. When comorbidities are present, the elderly seem to report better experiences than younger patients. Some differences in experience can be explained by sociodemographic and economic factors. No differences exist between conditions co-occurring with diabetes. Conclusion: Patients with diabetes who also suffer from other conditions report worse experiences than individuals who suffer from diabetes only. No specific conditions explain the differences in care experience.

## 1. Introduction

Diabetes mellitus is a chronic disease that is more prevalent among the elderly and recognized as an important public health problem in many countries, and is usually accompanied by multiple comorbidities [1]. Its burden is increasing, with an estimated prevalence in Europe among adults rising from 151 million in 2000 to 463 million in 2019, so that 1 in 11 adults have diabetes; 296,500 children and young people have type 1 diabetes, and these numbers are growing year to year and contributing to an increase in adult prevalence [2]. 

The Di@bet.es nationwide population-based cohort study in Spain showed an incidence of type 2 diabetes of 11.6 cases/1000 person-years, and an incidence of known diabetes of 3.7 cases/1000 person-years in Spain [3]. The prevalence of type 2 diabetes mellitus was 9.12% amongst all citizens aged ≥ 35 in the Basque Country in 2011 [4], and 10.6% according to a study published in 2017 [5]. This Basque Country prevalence rate (10.6% in 2017) is lower than the prevalence rate for Spain, which was estimated as 13.8%, according to results from a recent systematic literature review [6].

Diabetes is undiagnosed in 41% of cases, leaving these undiagnosed sufferers at risk of complications and resulting in higher healthcare costs [2]. Europe spends $161 billion (USD) of the total health budget on diabetes care [2]. In Spain, the mean cost for the healthcare system of treating a patient with diabetes aged between 20 and 79 years is estimated at $2651.5 (USD) [2]. 

Patient-centered approaches consider not only the clinical aspects of care, but also the patient’s experience [7,8], and, in patients with diabetes, are recognized for creating a high quality of care [9]. Patients’ experiences have been widely acknowledged to be an effective measure of the quality of healthcare delivery to patients with diabetes [9] and other chronic conditions [10]. The Chronic Care Model (CCM) has been shown to be an effective framework for improving the quality of diabetes care [11,12]. Better communication between patients and providers has been evidenced [11]. 

The Basque Country provides public healthcare to over two million inhabitants in the region, and around 18% are aged 65 or over [13]. The Basque Country has a Strategy for Tackling the Challenge of Chronicity [14]. It contains policies and projects aimed at reinventing the health delivery model with the purpose of improving the quality of care for chronic patients and advancing towards a more sustainable, proactive, and integrated model. The strategy includes plans to address different chronic problems, including secondary prevention and intensified follow-up of patients with diabetes. Diversity in Patient-Reported Experience Measures (PREM) still exists among patients with chronic conditions. A systematic literature review [15] showed that inequalities in healthcare experience exist among patients with diabetes by socioeconomic status (SES). Individuals from lower SES and more deprived areas are often found to have worse processes and intermediate outcomes indicators. Studies report opposing findings on patient experience with diabetes care according to sociodemographic characteristics and social and educational level [16,17]. A study [18] found that experiences of patients with multiple long-term conditions are not different from patients with a single long-term disease. However, multi-morbidity is generally associated in the literature with poorer health outcomes [19,20], and providers face extra challenges when managing these patients [21]. This inconclusive evidence shows that more research looking at population-based studies is needed. 

Few questionnaires validated to assess patients’ experiences cover all dimensions and concepts of healthcare comprehensively [22,23]. Many instruments do not include aspects relating to the development of information and communication technologies. The Basque Health Survey incorporates the Instrument for Evaluation of the Experience of Chronic Patients (IEXPAC) as a means for evaluating the experience of care of patients with chronic conditions. The IEXPAC instrument (available at http://www.iexpac.org, accesed on the 5 January 2021) [24] can detect differences between patient subgroups. In addition to information about patients’ clinical and risk factors, it has been shown to be an important piece of information for decision makers [25]. IEXPAC introduces a new focus on the interaction between patients and healthcare teams through the use of new technologies and patient-to-patient interactions [26]. It incorporates a broader notion of integrated care, including social care and patient self-management [26].

This article aims to determine whether there are differences in the experience of care amongst patients with a declared medical diagnosis of diabetes based on their sociodemographic, economic, and health-related characteristics.

## 2. Materials and Methods

### 2.1. The Basque Health Survey

The Basque Health Survey (ESCAV) is conducted by the Basque government every five years with people living in the Basque Country. The 2018 survey employed a representative sample of 5300 households and two questionnaires: (1) the family questionnaire, *n* = 12,995 individuals, and (2) the individual questionnaire, *n* = 8036. This sample of people surveyed is representative of the Basque Country population. The aims, methodology, and sampling are explained elsewhere (https://en.eustat.eus/document/encsalud_i.html, accessed on the 5 January 2021). 

### 2.2. Design and Working Sample

We conducted a cross-sectional study. Our dataset includes information from individuals who have declared having at least one type of chronic problem that has been diagnosed by a medical doctor. However, because our work focused on people with diabetes-related problems, our sample consisted only of those individuals who declared having at least a diabetes diagnosis. This created a working sample of *n* = 555 respondents aged 25 or older reporting diabetes and who live in the Basque Country. Data on the self-assessed experience of care was obtained from the Basque Health Survey 2018, which also included self-reported sociodemographic and economic data. 

### 2.3. Study Variables: Dependent Variables

We used the IEXPAC instrument to assess the experience of patients with chronic conditions. As dependent variables, we used the scores declared by our sample of patients with chronic conditions for three IEXPAC factors, derived from a combination of items from the 11 items of the IEXPAC questionnaire, and the global IEXPAC score. For each individual IEXPAC item, patients responded on a five-point Likert scale. The three factors (generated and provided with the dataset) and overall IEXPAC score take values between 0 and 10, a score of 0 representing the worst possible reported experience and a score of 10 representing the best possible reported experience. The three factors were: 

Factor (1) Productive Interactions (INTER): characteristics and content of interactions between patients and professionals oriented to improve outcomes; 

Factor (2) The New Relational Model (NEW): new ways of patient interaction with the healthcare system, through the internet or with peers; 

Factor (3) Patient self-management ability (SELF): the ability of individuals to manage their own care and improve their wellbeing based on professional-mediated interventions. 

Finally, the overall experience was measured by the IEXPAC score (OVERALL IEXPAC) that presents a summary of the patient’s experience with the whole healthcare delivery process. For a full description and meaning of each of the items in the IEXPAC, please refer to Figure 1.

### 2.4. Study Variables: Independent Variables

We explored sociodemographic and health-related characteristics.

(1) Sociodemographic and economic characteristics: gender, age, level of education, income, and occupation.

Gender is a dichotomic variable that takes a value of 1 if the individual is a man and 0 if the respondent is a woman. 

For age, we used the classification of the National Institute for Statistics in Spain. We split, however, the sample of respondents aged over 75 into two groups to test the specific effect of multi-morbidity among the elderly: respondents aged 75–90 and aged 90 or over. 

The level of education was categorized in the health survey as no education, primary education, lower secondary education, higher secondary education, and tertiary education. 

Income was categorized in income deciles of equivalent net household income. 

Occupation was categorized according to the National Classification of Occupations, which came into force in 2011 on the basis of the Social Determinants Working Group of the Spanish Society of Epidemiology’s [27] proposal. This classification groups occupational social classes into five groups: managers of companies of highly educated employees (Managers I), managers of companies of less educated employees (Managers II), intermediate occupations or freelancers (Intermediate), supervisors or technical position at qualified or semi-qualified occupations (semi-qualified), and supervisors or technical position at non-qualified occupations (Non-qualified). Note that occupation might be current (for people of working age) or past (for retired individuals), and the same categories apply for both types of respondents.

(2) Health-related variables include chronic conditions and the number of chronic health conditions. 

Chronic conditions. All 39 chronic conditions presented in the Basque Health Survey are included. Each disease is a dichotomous variable (takes a value of 0 or 1). For a disease to be included (value = 1), a patient has to declare that they have been diagnosed by a physician. The list of conditions included diabetes, which was used for the sample selection for this analysis. An individual can declare that they suffer from any amount of conditions on the list at the time of the survey. 

Number of chronic conditions. This variable was created based on the number of self-declared chronic conditions. We created ranges: one chronic disease (which has to be diabetes, given that our sample is selected as patients who have reported to suffer from diabetes), two chronic conditions (diabetes plus one more comorbidity), three chronic conditions (diabetes plus two more comorbidities), and more than three chronic conditions (diabetes plus three or more comorbidities).

### 2.5. Hypothesis

In this study, we tested the following hypotheses: If, in the Basque Country, there are inequalities in reported experiences of healthcare amongst people with chronic diabetes problems, according to the individual’s sociodemographic characteristics (such as gender, age, education, or occupation).If, in the Basque Country, and among people with diabetes, there are some chronic comorbidities that can be associated with worse reported experiences of healthcare than others.If, in the Basque Country, and among people with diabetes, those with multiple other chronic comorbidities report worse healthcare experiences than those with a lower number of chronic comorbidities.

### 2.6. Data Analysis

We started by conducting a descriptive analysis showing the distribution of patients’ responses and mean age for each of the IEXPAC items, IEXPAC factors, and overall IEXPAC score. We then performed a regression analysis using IEXPAC factors as dependent variables. 

We estimated the following linear regression models using ordinary least squares (OLS): IEXPAC_fi_ = α_0_ + α_k_ × X_i_ + ε_i_(1)
IEXPAC_fi_ = α_0_ + α_k_ × X_ki_ + α_j_ × CD_ji_ + ε_i_(2)
IEXPAC_fi_ = β_0_ + β_k_ × X_ki_ + β_(max k)_ + 1 × numberCD_i_ + ε_i_(3)
where (1) is the sociodemographic and economic characteristics model, (2) is the chronic conditions model, and (3) is the multi-morbidity model; f is a vector of dependent variables, the IEXPAC factors (INTER, NEW, and SELF), and the IEXPAC global score (OVERALL); X_i_ is the vector of k sociodemographic variables, included as dummies; CD_ji_ is the vector of j = 39 − 1 chronic conditions (as all respondents have declared having diabetes) for the i individual; numberCD_i_ is the number of chronic conditions for the i individual; and ε_i_ is the error term of the models.

Two- and three-way interactions between age, socioeconomic status (proxy by occupation), and education were included to test whether the effect of socioeconomic status varied between age groups. Interactions between the number of chronic conditions and (1) age ranges and (2) the number of chronic conditions or severity declared were also included to test if there were differences in the experiences of patients between age groups or those with multiple chronic conditions. Interactions between gender and other sociodemographic characteristics were tested, but produced insignificant effects of minimal added value, so these were omitted from the model. We used a confidence level of at least 95% in our analyses.

We tested and corrected the model for heteroscedasticity using heteroscedasticity-consistent standard errors (Eicker–Huber–White standard errors). This implies weighting the variances–co-variances matrix. This method, known as weighted least squares (WLS) makes the variance of the model robust and significantly reduces bias of heteroskedastic OLS estimators. We did not use any imputation method to replace missing data. Statistical analyses were conducted using Stata SE software.

## 3. Results

### 3.1. Descriptive Results

Among the 555 respondents declaring a diagnosis of diabetes, 46.3% were women. The mean age of the sample was 70.86 years old. None of the respondents were under the age of 28, and none reported a monthly net income over €5000. 

More than 70% of the patients responded “always” or “mostly” to the items that related to productive interactions (Factor 1: items 1, 2, 5, and 9), and self-management abilities (Factor 2: items 4, 6, 8, and 10), except for Factor 2, item 10, where the percentage was below 40%. 

For the items relating to the new relational model (Factor 3: items 3, 7, and 11), less than 20% responded “always” or “mostly”. These results are similar to the findings in a previously published study [28]. The global IEXPAC score mean was 5.92.

Table 1 and Table 2 below show the distribution of responses by sociodemographic, economic, and health-related variables, as well as the mean age and mean values of IEXPAC factors (and standard deviation—s.d.) and global scores for each variable subgroup or category. The mean for each variable is provided together with its standard deviation. Given that we have converted all variables into categorical variables, the mean value of a certain category can also be interpreted as the proportion of respondents in that specific category. For example, the mean for women was 0.463, and the mean for men was 0.537. The sum of both means equaled 1, which means that 0.463 was the proportion of women (or 46.3%) with respect to the total number of respondents in our study population. Figure 1 shows the distribution of responses (in percentage of patients) for each of the IEXPAC items.

The distribution of respondents by declared conditions and by the number of declared conditions is also shown. Conditions are sorted in this table according to prevalence. 

The condition with the highest mean age was dementia (*n* = 22, 80.45 years old). The most frequently declared conditions in this sample of patients with self-declared diabetes were hypertension (*n* = 338) and cholesterol (*n* = 295), and these two conditions affected 11.41% of the respondents, plus rheumatisms *(n* = 104). 

The low mean experience score with the new relational model indicated that our respondents reported a bad experience with this factor compared to the rest of the factors and to the global IEXPAC score. This is a significant result, and one which we discuss further in Section 5. 

### 3.2. Regression Results

Results from the estimation of Model 1 are shown in Table 3 below (a detailed version of the table, Appendix A, including interaction effects and 95% confidence intervals, is available as online supplementary Materials). 

Better experience is associated with being older, except for the very old respondents aged 90 or over. For example, for respondents aged 64–75, experience with factors related to self-management was 3.826 points higher in the mean than for respondents aged 25–44, with a *p*-value < 0.01. 

However, when we interacted age with occupation (See Appendix A in the Appendix A), better experience was associated with being younger when the respondent worked in a qualified or semi-qualified position. A similar result was observed when we compared experiences with services oriented to improve self-management abilities between older individuals and younger individuals in non-qualified occupations. The coefficients for the interaction effects were negative, but not large enough to compensate for the positive coefficients for the age ranges. The opposite was observed when we looked at overall experience. In this case, for those aged 64–75 and reporting a non-qualified occupation, the mean score for global experience of utilization of healthcare services would be 4.351 + 1.983 − 2.073 = 4.261 (note that −2.073 was the effect of reporting non-qualified occupations and being aged 64–75. To see the estimates of all interaction effects included in the regression model, please see the provided supplementary material). This was, according to our model estimates, a lower experience score than an individual aged 25–44 would report. We also found that respondents aged 75–89 and who had secondary/upper education had better experiences with the new relational model than younger individuals with primary education (for respondents aged 75–89, coef. = 3.248, *p*-value < 5%). Similar results were observed for respondents aged 90 or over.

Results from Model 2 are shown in Table 4 (a detailed version of the table, Appendix A, including interaction effects and 95% confidence intervals, is available as online Appendix A).

Patients with constipation had a significantly reduced patient experience for two factors (productive interactions and self-management) and with the overall experience. A diagnosis of “other mental” also appeared to significantly and negatively affect experiences with the new relational model compared to other conditions. These were the only two conditions that had significant effects at the 95% level. One should note, however, that constipation and other mental conditions were only reported by a very small proportion of respondents, and therefore, their impact on the population experience levels will be small.

Model 3 results are shown in Table 5 (a detailed version of the table, Appendix A, including interaction effects and 95% confidence intervals, is available as online Appendix A). Most confounding factors led to experience differences across all IEXPAC factors and for the global value. There was consistency with previous results found in Models 1 and 2 for confounding factors. Additional effects arose when we introduced the number of conditions as an additional covariate, as well as interactions between the number of conditions and the age of the respondents. 

There was a significant difference in the valuation of experience between patients having multiple conditions and patients with diabetes only. Having two chronic conditions decreased experience with the factor related to improvement of self-management abilities by 11.37 points on average. This impact was higher compared to the impact of having diabetes only (the mean experience for patients with diabetes only would be 13.47 − 11.37 = 2.1 points of the IEXPAC). However, if, in addition to having two conditions, we knew the respondent was aged 64–75 instead of 25–44 (the base age group in our estimation), we would predict their experience with the same factor (SELF) to be 13.47 − 5.607 − 11.37 + 11.60 = 8.093. Note that + 11.60 referred to the interaction effect of being aged 64–75 and declaring two chronic problems (see Appendix A in the Appendix A). Age had a significant impact when it interacted with the number of comorbidities. For all factors and for the overall IEXPAC score, the size of the effect of the number of conditions plus the interaction effect between age and the number of conditions was smaller than the size of the effect of age itself. The result was always negative, indicating worse experiences for younger respondents. This was true for all age ranges and in all four models. 

## 4. Study Strengths and Limitations

Our study’s main strength is that it is the first study that analyzes patients’ evaluations of their experiences with healthcare services across several factors, using a sample with prevalence rates of chronic conditions that is representative of the wider population. Despite the fact that chronic problems were not used to ensure representativeness of the population in the ESCAV 2018 survey (the full ESCAV18 methodology report can be consulted at: https://www.euskadi.eus/contenidos/informacion/enc_salud_18_metodologia/es_def/adjuntos/Metodologia-encuesta-salud-2018.pdf, accessed on the 5 January 2021), the prevalence rates of our participants who declared a diagnosis by a doctor of a chronic problem are almost identical to prevalence rates of patients who have a clinical diagnosis of chronic problems in the Basque Country. Therefore, we can assume that their experiences should be representative of the wider population of diabetes sufferers in the Basque Country. 

The study does suffer from some limitations. First, reported chronic conditions are self-declared, although our evidence from diabetes suggests that these types of self-declared reports, where the question relies on a declaration of a diagnosis confirmed by a doctor, are reliable. Second, we count conditions to control for multi-morbidity, but there is no information regarding the severity or the progression of the conditions. Third, we analyzed IEXPAC factors instead of conducting the analyses for each of the IEXPAC items, and, therefore, we could be missing important information if there are differences between the distributions of responses in each factor. However, our descriptive analyses showed almost identical results to the previous literature, as explained below in the discussion. Fourth, other potential variables, such as patient ethnicity or the health area responsible for providing care, which would be desirable confounding factors to include, were not available. Finally, the IEXPAC questionnaire included in the ESCAV survey asks questions regarding experience with healthcare services, but we cannot derive from responses if these are public or private services. Possible explanations and implications for these limitations are presented in the Discussion section, along with suggestions for further research. 

## 5. Discussion

This paper presents descriptive and regression analyses on the experience of care using the IEXPAC amongst patients declaring diabetes in the ESCAV Survey 2018 in the Basque Country. 

Our analysis adds to the existing literature in various ways. First, we have data from a survey of a representative population, filtered for those declaring chronic diabetes. Secondly, these subjects receive care from a multidisciplinary care team.

In addition to finding differences in the experiences of patients with diabetes along sociodemographic and economic lines, we found that patients with diabetes who suffer from additional comorbidities reported worse experiences overall, but also worse levels of experience with all IEXPAC factors, than individuals who suffer from diabetes only. Specific conditions do not appear in our analysis to be relevant in explaining differences in care experience, except for a small number of conditions. 

The first limitation of this study is that, for identification of chronic problems, including diabetes, we are relying on self-declared diagnoses. Model 2 estimates (the conditions model) show that there are no significant differences found in how patients with different conditions value experience with the utilization of healthcare services. We argue that this shows that the number of comorbidities is more important than the specific disease in this population. Our second limitation refers to the use of IEXPAC factors as dependent variables for the regression analysis instead of IEXPAC independent items (several IEXPAC items form each of the factors, and each item is included in one factor only). We have conducted descriptive statistics for each IEXPAC item (see Figure 1), and our results were very similar to those of a previous study, which conducted a descriptive analysis using IEXPAC items, but then used IEXPAC factors as dependent variables for the regression analysis [28]. Although the context of that study was not the same—it was conducted with a non-representative sample of patients with inflammatory bowel disease, and used the aggregated factors for the multivariate analyses—their analysis gives us confidence that this approach is valid and should give unbiased, relevant results. Given that the sample surveyed is designed to be representative of the wider population, we do not suspect that omitting variables, such as ethnicity and health area, will bias our analysis. Finally, we address the lack of information regarding the use of private or public providers by noting that individuals’ accessibility to private healthcare services could be proxied by income or occupation. In our sample, the majority of respondents belong to the lowest SES groups and these respondents are unlikely to have private insurance. 

To our knowledge, this is the first population-based study using a validated instrument for the measurement of patient experience that disaggregates experiences across several dimensions in an integrated care delivery system. Similar research using medical diagnosis and studying long-term, life-limiting chronic condition populations is encouraged. 

We identified patterns consistent with previously published literature. 

First, our population with self-declared diabetes reported on average good levels of patient experience, with high mean scores in all IEXPAC factors, except the factor that relates to the new relational model. This is in line with previous evidence. Studies have found that people with diabetes report better experiences of healthcare services than people without diabetes [29]. This could have a behavioral explanation. 

Our population of respondents may not have that much of a choice, and even if they do have a choice, they usually do not exert it. Chronically ill patients often prefer, for example, the nearest hospital simply for convenience [30], even if it is not the best hospital in their hospital choice set. They may not even be aware that there could be better care available if they travelled further. Additionally, with a chronic condition such as diabetes, the consequences of not receiving any treatment are worse than the consequences of receiving poor care, and, therefore, diabetes patients may be more accepting of customer service shortcomings than patients with other problems, where bad customer service may play a more important role in the patients’ assessment of their care. 

Our descriptive analysis reveals that most of the individuals in our study belong to lower SES groups. We specifically found that younger patients with diabetes who are/were in qualified or semi-qualified occupations have a better experience of care. A recent study found that the healthcare experience among people with diabetes varies by sociodemographic group [29]. In addition, a review study also found that individuals from low SES groups have a higher future risk of developing diabetes, although this study reviewed results for obese populations only [31]. However, although studies have analyzed the effect of low SES (including occupation as an explanatory factor of SES) on satisfaction with healthcare services of diabetes patients [32], there is literature that has demonstrated that satisfaction scores present a limited and optimistic picture compared to experience scores [33]. Our paper is, thus, making a contribution in this aspect, which has not previously been assessed in a population-based study. 

Third, we confirmed that multi-morbidity has a direct impact in poorer diabetes care experience for aspects relating to access and communication in primary care, as in previous publications [29]. This is true in our study with and without interacting the number of comorbidities with age groups, which has also been found in previous publications [17,22]. We found that, among people with self-declared diabetes, those with higher numbers of additional comorbidities reported worse primary care experiences than those with lower numbers of additional comorbidities, reinforcing findings by other studies in which individuals with poorer health report worse primary care experiences [29]. 

Amongst sociodemographic characteristics, the most important is age. This result may be conditioned by the higher scores observed for the new relational model factor for younger patients, which could be showing differences in the use of digital information technologies between the oldest and younger groups and those with lower versus higher education, as observed in previously published studies [34]. Younger patients with secondary upper levels of education show higher scores of experience of care, a result that is also consistent with previously observed patterns [16]. We also observe lower rates for items relating to the new relational model, similar to other studies in Spain that use the IEXPAC [24,26]. 

We did not find significant differences in patients’ experiences across the analyzed IEXPAC factors by gender. However, other studies have found contradictory findings with respect to the influence of gender on healthcare experiences [35,36,37], suggesting the need for further study of this topic. In addition, our analysis did not return statistically significant variations in the reported experience of care according to different chronic conditions, except for patients with constipation (in addition to diabetes) and other mental health conditions. These, however, were reported only by a small proportion of respondents, so their impact on the experience at the population level is small. Recently, in a study that computed the odds ratios for reporting a good experience of care among the population with diabetes and other comorbidities and the population without diabetes, a similar result was observed [29]. 

We believe that this paper contributes to a better understanding of the effect of socio-demographic, economic, and health-related characteristics on patients’ care experiences. The lower levels of satisfaction reported with the new relational model reflect the need of strategies that can help to harness the opportunities that new technologies and internet resources present for quality of care improvement [38,39,40]. Research using diagnostic information is encouraged in order to contrast our results with those from a clinically diagnosed population, disaggregating by the type of diabetes where possible. 

## 6. Conclusions

In conclusion, in this study of a sample representative of the Basque Country population, we find that experience with the utilization of healthcare services in people with diabetes is more negatively affected by the number of comorbidities than by a diagnosis of any specific disease. Our population, people living in the Basque Country with at least declared chronic diabetes, overall reports good levels of experience with most of the analyzed aspects of the healthcare received. However, we also observed lower experience ratings with the new relational model factors, which suggests that there is still work to do on improving the new technologies and internet resources offered to the population we have studied. 

## Figures and Tables

**Figure 1 healthcare-09-00509-f001:**
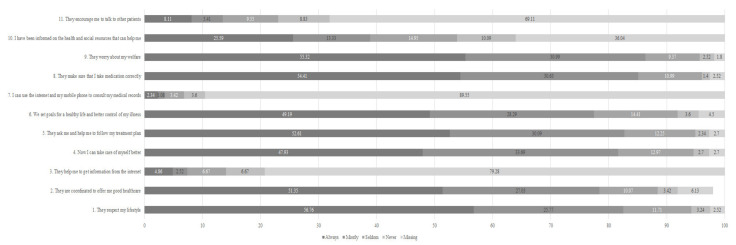
Distribution of patients’ responses to IEXPAC items. Numbers in bars represent the percentage of respondents who responded to each option.

**Table 1 healthcare-09-00509-t001:** Sociodemographic and economic variables. Descriptive statistics.

Variables	Category	Obs.	Mean (s.d)	Age Mean	Factor 1: INTER Mean (s.d)	Factor 2: NEW Mean (s.d)	Factor 3: SELF Mean (s.d)	OVERALL IEXPAC Mean (s.d)
Gender	Women	257	0.463 (0.499)	73.01	8.200 (2.191)	1.008 (1.737)	7.230 (2.126)	5.886 (1.661)
	Men	298	0.537 (0.499)	69.01	8.154 (1.920)	1.407 (2.080)	7.160 (2.052)	5.953 (1.658)
Age	25 to 44	13	0.023 (0.151)	39.38	7.596 (3.469)	2.244 (3.717)	5.817 (3.108)	5.490 (2.915)
	45 to 64	137	0.247 (0.432)	57.89	7.870 (2.278)	1.478 (2.021)	7.030 (2.066)	5.821 (1.729)
	64 to 75	184	0.332 (0.471)	70.02	8.179 (1.868)	1.259 (1.913)	7.262 (1.899)	5.958 (1.566)
	75 to 89	207	0.373 (0.484)	80.73	8.400 (1.867)	1.002 (1.759)	7.310 (2.123)	5.986 (1.577)
	90 +	14	0.025 (0.157)	92	8.348 (2.613)	0.536 (0.959)	7.411 (2.639)	5.877 (1.922)
Net monthly income	No income	1	0.003 (0.058)	45	6.250 (.)	0 (.)	6.250 (.)	4.545 (.)
	0 to €500	2	0.007 (0.081)	61	8.750 (0.884)	1.250 (0.589)	7.500 (0.884)	6.250 (0.482)
	€501 to €1,000	82	0.272 (0.445)	74.75	8.255 (1.808)	0.996 (1.975)	7.416 (1.944)	5.970 (1.532)
	€1,001 to €1,500	106	0.351 (0.478)	73.14	8.325 (1.916)	0.896 (1.426)	7.105 (1.839)	5.855 (1.440)
	€1,501 to €2,000	49	0.162 (0.369)	71.26	8.010 (2.492)	1.071 (1.623)	7.041 (2.420)	5.765 (1.878)
	€2,001 to €2,500	32	0.106 (0.308)	66.62	8.086 (2.256)	2.370 (2.744)	7.168 (2.454)	6.193 (2.042)
	€2,501 to €3,500	20	0.066 (0.249)	64.25	8.344 (1.883)	1.625 (2.955)	7.500 (1.298)	6.205 (1.568)
	€3,501 to €5,000	10	0.003 (0.179)	62.3	9 (1.508)	1.917 (2.153)	7.313 (1.445)	6.455 (1.199)
	Missing responses	253	-	-	-	-	-	-
Level of education	Primary	264	0.476 (0.5)	75.15	8.333 (2.049)	1.124 (1.753)	7.320 (2.114)	5.999 (1.627)
	Secondary—lower	135	0.243 (0.429)	69.47	7.958 (1.906)	0.877 (1.764)	6.986 (1.909)	5.673 (1.538)
	Secondary—upper	114	0.205 (0.404)	65.36	8.059 (2.104)	1.513 (2.073)	7.198 (2.156)	5.961 (1.758)
	Tertiary (university)	42	0.076 (0.265)	63.28	8.199 (2.307)	2.163 (2.718)	7.039 (2.247)	6.131 (1.904)
Occupation	Managers I	68	0.123 (0.328)	71.13	8.327 (1.834)	1.434 (2.225)	7.123 (2.066)	6.009 (1.577)
	Managers II	29	0.052 (0.223)	65.37	8.470 (2.022)	1.954 (2.628)	7.866 (1.937)	6.473 (1.724)
	Intermediate	26	0.047 (0.212)	71.96	7.788 (2.188)	0.865 (1.384)	6.971 (1.756)	5.603 (1.512)
	Semi-qualified	130	0.235 (0.424)	70.72	8.327 (1.998)	1.404 (1.988)	7.462 (2.240)	6.124 (1.672)
	Non-qualified	301	0.543 (0.499)	71.28	8.075 (2.107)	1.060 (1.790)	7.037 (2.043)	5.785 (1.664)
	Missing responses	1	-	-	-	-	-	-

**Table 2 healthcare-09-00509-t002:** Health-related variables: diabetes and comorbid conditions. Descriptive statistics.

Variables	Obs.	Mean (s.d)	Age Mean	Factor 1: INTER Mean(s.d)	Factor 2: NEW Mean(s.d)	Factor 3: SELF Mean(s.d)	OVERALL IEXPAC Mean (s.d)
Diabetes	555	1 (0)	70.86	8.176 (2.048)	1.222 (1.937)	7.193 (2.085)	5.922 (1.658)
Hypertension	338	0.609(0.488)	71.69	8.225 (1.934)	1.223 (1.980)	7.163 (2.005)	5.929 (1.600)
Cholesterol (high)	295	0.532 (0.499)	71.91	8.169 (1.898)	1.093 (1.887)	7.085 (1.973)	5.845 (1.556)
Rheumatisms	104	0.187 (0.391)	72.96	8.005 (2.151)	0.601 (1.264)	7.019 (2.127)	5.627 (1.575)
Other, heart	91	0.164 (0.371)	77.27	8.283 (1.908)	1.172 (1.968)	7.308 (2.101)	5.989 (1.632)
Lower back pain	84	0.151 (0.359)	73.20	7.939 (2.035)	0.516 (1.260)	6.830 (1.954)	5.511 (1.458)
Blood circulation	65	0.117 (0.322)	74.16	7.894 (2.158)	0.923 (1.635)	6.856 (2.114)	5.615 (1.627)
Upper back pain	59	0.106 (0.309)	73.15	7.977 (1.699)	0.424 (1.141)	6.769 (1.741)	5.478 (1.280)
Thyroids	56	0.101 (0.301)	71.03	8.002 (2.367)	1.235 (1.946)	7.121 (2.502)	5.836 (1.878)
Insomnia	54	0.097 (0.297)	73.44	7.535 (2.539)	0.602 (1.396)	6.655 (2.348)	5.324 (1.816)
Deafness	49	0.088 (0.284)	77.08	7.844 (1.977)	0.527 (1.223)	6.888 (2.047)	5.501 (1.471)
Varicose veins (in legs)	40	0.072 (0.259)	75.47	8.297 (1.343)	0.708 (1.355)	7.094 (1.617)	5.790 (1.171)
Osteoporosis	34	0.061 (0.24)	75.17	7.831 (2.231)	0.417 (1.010)	6.397 (2.399)	5.287 (1.575
Other, mouth	32	0.058 (0.233)	71.87	7.754 (1.760)	0.573 (1.187)	6.719 (1.817)	5.419 (1.127)
Kidney	31	0.056 (0.23)	74.67	7.581 (2.414)	0.941 (1.502)	6.573 (2.376)	5.403 (1.755)
Asthma	30	0.054 (0.226)	71.63	7.875 (2.609)	0.972 (2.178)	7.083 (2.420)	5.705 (2.060)
Incontinence	28	0.05 (0.219)	77.67	7.813 (2.676)	0.744 (1.625)	6.473 (2.991)	5.398 (2.058)
Prostatitis	27	0.049 (0.215)	76.33	8.426 (1.495)	0.957 (1.510)	7.083 (2.080)	5.901 (1.448)
Anxiety	27	0.049 (0.215)	67.77	7.824 (2.168)	1.080 (2.141)	6.597 (2.472)	5.539 (1.874)
Depression	26	0.047 (0.212)	69	8.053 (1.493)	0.609 (1.015)	7.404 (1.548)	5.787 (1.003)
Other chronic	25	0.045 (0.208)	68.04	8.525 (1.857)	1.233 (1.786)	7.775 (1.615)	6.264 (1.307)
Cataracts	24	0.043 (0.204)	77.41	8.203 (1.156)	0.764 (1.610)	7.057 (1.772)	5.758 (1.255)
Hemorrhoids	23	0.041 (0.199)	72.60	7.500 (2.230)	0.725 (1.672)	6.495 (2.283)	5.287 (1.653)
COPD	22	0.04 (0.195)	69.86	7.642 (2.873)	0.720 (1.895)	7.074 (2.480)	5.548 (2.112)
Dementia	22	0.04 (0.195)	80.45	8.182 (2.609)	1.250 (2.004)	6.648 (3.299)	5.733 (2.302)
Blindness	20	0.036 (0.187)	70.85	7.188 (3.004)	1.208 (1.587)	6.844 (2.963)	5.432 (2.240)
Cancer	19	0.034 (0.182)	72.73	8.158 (1.892)	1.447 (1.775)	7.336 (1.919)	6.029 (1.613)
Constipation	19	0.034 (0.182)	75.73	7.336 (1.941)	0.658 (1.536)	6.053 (2.073)	5.048 (1.520)
Caries	18	0.032 (0.177)	71.61	7.743 (2.124)	0.694 (1.572)	6.736 (2.274)	5.455 (1.699)
Migraine	18	0.032 (0.177)	75.05	7.361 (2.595)	0.046 (0.196)	5.729 (2.870)	4.773 (1.884)
Anemia	18	0.032 (0.177)	74.5	8.611 (1.146)	0.880 (1.750)	7.257 (1.341)	6.010 (1.002)
Thrombosis	17	0.031 (0.172)	71.11	7.904 (2.450)	0.980 (2.046)	6.618 (2.370)	5.548 (1.905)
Skin	16	0.029 (0.167)	76.5	7.578 (1.751)	0.677 (1.528)	6.914 (1.847)	5.455 (1.473)
Acute Myocardial Infarction	14	0.025 (0.157)	74.85	7.902 (1.828)	0.714 (1.217)	6.964 (1.605)	5.601 (1.042)
Stomach ulcer	13	0.023 (0.151)	65.92	7.260 (3.457)	1.282 (2.247)	5.817 (3.221)	5.105 (2.716)
Allergy	9	0.016 (0.126)	63.11	7.014 (3.708)	0.648 (1.085)	5.903 (3.064)	4.874 (2.574)
Oher, mental	9	0.016 (0.126)	67	7.292 (1.952)	0.463 (0.735)	6.597 (2.690)	5.177 (1.618)
Diabetic foot	8	0.014 (0.119)	75.625	7.734 (1.668)	1.771 (2.417)	7.109 (2.586)	5.881 (1.888)
Fibromyalgia	7	0.013 (0.112)	63.57	8.214 (1.990)	1.667 (1.596)	7.411 (0.983)	6.136 (0.964)
Number of chronic conditions							
1	58	0.105 (0.306)	63.84	8.351 (1.965)	1.250 (1.901)	7.198 (2.150)	5.995 (1.604)
2	114	0.205 (0.404)	69.38	8.163 (2.135)	1.740 (2.171)	7.495 (2.031)	6.168 (1.781)
3	128	0.231 (0.422)	71.76	8.071 (2.115)	1.361 (2.153)	7.090 (2.143)	5.884 (1.744)
> 3	255	0.459 (0.499)	72.67	8.194 (2.002)	0.915 (1.654)	7.108 (2.063)	5.814 (1.564)

All chronic conditions and the number of chronic conditions are dummies, taking values 0 or 1; COPD: chronic obstructive pulmonary disease; all variable subgroups are dummy variables, taking values 0 or 1. Therefore, the variable mean can easily be converted (% = variable mean × 100) into the percentage of individuals in each subgroup/category.

**Table 3 healthcare-09-00509-t003:** Model 1—WLS results. Differences in healthcare experience among patients with self-declared diabetes. The effect of sociodemographic and economic characteristics.

Variable	Category	Factor 1: INTER	Factor 2: NEW	Factor 3: SELF	OVERALL IEXPAC
Gender. Baseline: Women	Men	−0.078	0.231	−0.151	−0.020
Age ranges. Baseline: 25–44	45 to 64	0.687	1.714	3.174 **	1.872 *
	64 to 75	0.791	1.114	3.826 ***	1.983 **
	75 to 89	0.424	0.805	3.305 **	1.576
	90 or over	−0.410	−0.131	1.100	0.215
Occupation. Baseline: Managers I	Managers II	2.141 **	5.555 ***	5.597 **	4.329 **
	Intermediate	0.200	−0.599	0.103	−0.053
	Semi-qualified	−1.655	5.356 **	−1.132	0.447
	Non-qualified	1.203 **	0.972 **	2.784 ***	1.715 ***
Education. Baseline: Primary	Secondary-lower	−3.149	2.461	−2.252	−1.293
	Secondary-upper	−1.962	−2.410	−0.688	−1.621
	Tertiary	−0.051	0.981	0.503	0.432
Constant		7.988 ***	−0.100	4.051 ***	4.351 ***
Interactions					
Occupation # Age ranges	YES	YES	YES	YES	YES
Education # Age ranges	YES	YES	YES	YES	YES
Goodness-of-fit	R-squared	0.067	0.092	0.074	0.058
	BIC	2536.062	2453.380	2550.006	2307.144
Sample size (¥)		554	554	554	554

* *p* < 0.1, ** *p* < 0.05, *** *p* < 0.001; BIC: Bayesian information criterion; the presented model is corrected from heteroscedasticity using Eicker–Huber–White standard errors. A detailed table including results for the interactions and 95% confidence intervals (95% CI) is available as Supplementary Material (Appendix A). ¥: There is one missing response for occupation, and this has been excluded from the analysis.

**Table 4 healthcare-09-00509-t004:** Model 2—WLS results. Differences in healthcare experience among patients with self-declared diabetes. The chronic conditions model.

Variable	Category	Factor 1: INTER	Factor 2: NEW	Factor 3: SELF	OVERALL IEXPAC
Chronic conditions	Hypertension	0.067	0.162	−0.069	0.044
	Cholesterol (high)	−0.038	−0.079	−0.197	−0.107
	Rheumatisms	0.213	−0.297	0.359	0.127
	Other, heart	0.112	0.188	0.261	0.187
	Lower back pain	−0.008	−0.397	−0.013	−0.116
	Blood circulation	−0.134	−0.003	0.005	−0.048
	Upper back pain	0.060	−0.477	−0.201	−0.181
	Thyroids	−0.126	0.270	0.127	0.074
	Insomnia	−0.594	−0.170	−0.329	−0.382
	Deafness	−0.195	−0.212	0.007	−0.127
	Varicose veins (in legs)	0.239	0.091	0.024	0.121
	Osteoporosis	−0.124	−0.447	−0.731	−0.433
	Other, mouth	−0.040	0.088	−0.113	−0.032
	Kidney	−0.743	−0.148	−0.713	−0.570
	Asthma	−0.121	−0.177	0.209	−0.016
	Incontinence	0.032	−0.337	−0.351	−0.208
	Prostatitis	0.148	−0.440	−0.177	−0.130
	Anxiety	0.256	0.670	−0.045	0.259
	Depression	0.175	−0.454	0.480	0.115
	Other chronic	0.567	−0.120	0.467	0.343
	Cataracts	−0.144	0.053	−0.191	−0.108
	Hemorrhoids	−0.311	0.107	−0.255	−0.177
	COPD	−0.432	−0.512	−0.049	−0.314
	Dementia	0.908	0.642	0.405	0.652
	Blindness	−0.997	0.023	−0.306	−0.468
	Cancer	−0.233	0.193	0.035	−0.019
	Constipation	−1.161 **	−0.142	−1.423 **	−0.978 **
	Caries	0.500	0.098	0.576	0.418
	Migraine	−0.747	−0.485	−1.164	−0.827
	Anemia	0.873 *	−0.144	0.418	0.430
	Thrombosis	−0.127	0.029	−0.252	−0.130
	Skin	−0.195	0.150	0.283	0.073
	AMI	−0.252	−0.273	−0.084	−0.197
	Stomach ulcer	−0.468	−0.373	−1.035	−0.648
	Allergy	−0.608	−0.524	−0.572	−0.572
	Other, mental	−0.862	−0.919 **	−0.443	−0.725
	Diabetic foot	−0.326	1.163	0.181	0.265
	Fibromyalgia	0.477	1.190 *	0.312	0.611
	Constant	8.546 ***	−0.560	3.770 **	4.326 ***
Goodness of fit	R-squared	0.124	0.147	0.140	0.123
	BIC	2740.861	2664.738	2749.151	2507.772
Sample size (¥)	N	554	554	554	554

AMI: Acute myocardial infarction; COPD: chronic obstructive pulmonary disorder; * *p* < 0.1, ** *p* < 0.05, *** *p* < 0.001; BIC: Bayesian information criterion; the presented model is corrected from heteroscedasticity using Eicker–Huber–White standard errors. A detailed table including results for the interactions and 95% confidence intervals (95% CI) is available as Appendix A (Appendix A). ¥: There is one missing response for occupation, and this has been excluded from the analysis.

**Table 5 healthcare-09-00509-t005:** Model 3—WLS results. Differences in healthcare experience among patients with diabetes. The effect of the presence of multi-morbidity.

Variable	Category	Factor 1: INTER	Factor 2: NEW	Factor 3: SELF	OVERALL IEXPAC
Gender. Baseline: Women	Men	−0.158	0.141	−0.223	−0.100
Age ranges. Baseline: 25 to 44	45 to 64	−7.575 ***	−8.357 ***	−6.026 ***	−7.225 ***
	64 to 75	−7.894 ***	−8.669 ***	−5.607 **	−7.274 ***
	75 to 89	−8.445 ***	−9.081 ***	−6.145 **	−7.782 ***
	>= 90	0.255	−3.717 ***	−1.244	−1.374
Education. Baseline: Primary	Secondary-lower	−10.690 ***	−5.331 **	−11.017 ***	−9.347 ***
	Secondary-upper	−10.876 ***	−11.054 ***	−10.678 ***	−10.853 ***
	Tertiary	−0.266	−0.653	−0.710 ***	−0.533
Occupation Baseline: Managers I	Managers II	5.632 **	8.280 ***	8.836 ***	7.519 ***
	Intermediate	0.174	−0.754	−0.004	−0.143
	Semi-qualified	4.364 **	8.130 ***	4.296 **	5.366 ***
	Non-qualified	3.020 *	1.102 **	4.466 **	3.023 **
Number of conditions. Baseline: 1	2	−12.073 ***	−10.657 ***	−11.372 ***	−11.432 ***
	3	−11.336 ***	−10.146 ***	−12.627 ***	−11.481 ***
	+3	−9.461 ***	−5.979 **	−7.002 ***	−7.617 ***
Interactions					
Occupation # Age ranges	YES	YES	YES	YES	YES
Education # Age ranges	YES	YES	YES	YES	YES
Number of conditions # Age ranges	YES	YES	YES	YES	YES
Constant		16.865 ***	9.556 ***	13.469 ***	13.636 ***
		(13.512,20.217)	(5.502,13.610)	(10.230,16.707)	(10.832,16.441)
Goodness-of-fit	R-squared	0.128	0.156	0.120	0.123
	BIC	2580.621	2488.809	2604.232	2343.248
Heteroscedasticity correction method	YES	Robust variance	Robust variance	Robust variance	Robust variance
	N	554	554	554	554

* *p* < 0.1, ** *p* < 0.05, *** *p* < 0.001; BIC: Bayesian information criterion; the presented model is corrected from heteroscedasticity using Eicker–Huber–White standard errors. A detailed table including results for the interactions and 95% confidence intervals (95% CI) is available as Supplementary Material (Appendix A). ¥: There is one missing response for occupation, and this has been excluded from the analysis.

## Data Availability

The datasets generated and/or analyzed during the current study are available from the corresponding author on reasonable request.

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
