# Peer review of "Factors Influencing Healthcare Experience of Patients with Self-Declared Diabetes: A Cross-Sectional Population-Based Study in the Basque Country"

_healthcare, 2021, doi:10.3390/healthcare9050509_

Round 1

Reviewer 1 Report

The manuscript submitted for publication by Nuño-Solínis et al., titled: “Factors influencing healthcare experience of patients with self-declared diabetes: a cross-sectional population-based study in the Basque Country”, is aiming to evaluate the healthcare experience of various diabetic patients in the Basque Country in regard to their sociodemographic/socioeconomic status. The manuscript is well written overall and addresses an important topic which becomes even more relevant  within the current healthcare systems.

Some suggestions and points that need to be addressed according to the reviewer’s assessment are the following:

  1. Line 41: Cost of diabetes per person: is this cost a one-time all cost? How is it appropriated?
  2. Could the authors provide some brief background information about the Basque population? It is important to clarify what are the main characteristics and/or special characteristics of the population.
  3. Consider providing a map identifying the location/geography pertinent to the studied population.
  4. The studied sample of 555 participants appears to cross from adolescent to adulthood. How do authors correct for this age crossover, especially when there is both underage and adult participants?
  5. What were the inclusion and exclusion criteria for the selection of the 555 cases studied? (i.e.: How were the 555 selected?).
  6. How was the number 555 decided? Was there a power calculation or another method that led to the selection of 555 out of the total number of datapoints available in the database?
  7. What were confounding factors and how were they controlled for (age, sex, body composition, BMI, years diagnosed with diabetes, smoking status etc) ?
  8. Please include a short section on strengths and limitation (limitations are actually mentioned briefly in the Discussion section) of the study.
  9. While the repository from which the authors drew cases is rather large the 555 sample is not necessarily deemed “corresponding/characteristic/representative” if the entire Basque population. What is the Basque population?
  10. Please provide a section with brief description of the study which repository was used to select the cases studied in addition to the reference (website).
  11. Line 187: please explain/correct “Error! Reference source not found”
  12. The discussion section is rather short and most importantly does not discuss the findings in relation to other similar studies or other populations.
  13. Could the results be interpreted differently? Is there a behavioral or social theory that could explain the results?
  14. The healthcare system through which care was received was it uniform? Were there patients receiving care from private versus public clinics? Was the accessibility financial, geographical, physical, cultural, race/ethnicity factors etc considered? All these parameters can impact the level of stress and anxiety and as a result the rating of the healthcare experience.
  15. Please include a formulated and clearly articulated hypothesis (Ho).
  16. The manuscript has a relatively limited list of references. More studies could be considered and cited. Some works to consider:

Sikalidis AK, Öztağ M Optimized snacking is positively associated with socioeconomic status and better Type 2 Diabetes Mellitus management in Turkish patients. Gazz Med Ital - Arch Sci Med. 2020; 179(7-8):459-67. doi:10.23736/S0393-3660.19.04159-9.

Volaco A, Cavalcanti AM, Filho RP, Précoma DB. Socioeconomic Status: The Missing Link Between Obesity and Diabetes Mellitus? Curr Diabetes Rev. 2018;14(4):321-326. doi: 10.2174/1573399813666170621123227.

Yin T, Yin DL, Xiao F, Xin QQ, Li RL, Zheng XG, Yang HM, Wang LH, Ding XY, Chen BW. Socioeconomic status moderates the association between patient satisfaction with community health service and self-management behaviors in patients with type 2 diabetes: A cross-sectional survey in China. Medicine (Baltimore). 2019 May;98(22):e15849. doi: 10.1097/MD.0000000000015849.

Author Response

Responses to Reviewer 1

The manuscript submitted for publication by Nuño-Solínis et al., titled: “Factors influencing healthcare experience of patients with self-declared diabetes: a cross-sectional population-based study in the Basque Country”, is aiming to evaluate the healthcare experience of various diabetic patients in the Basque Country in regard to their sociodemographic/socioeconomic status. The manuscript is well written overall and addresses an important topic which becomes even more relevant within the current healthcare systems.

Thank you for the time you devoted to read, comment and improve our paper. We appreciate your positive critique of the paper. We believe that your comments were relevant and essential to improve the quality of the paper. In addition, please note that the resubmitted version of the manuscript includes the English revision suggested. 

Some suggestions and points that need to be addressed according to the reviewer’s assessment are the following:

We now respond, one by one, to each of your comments. Please, note that when referring to specific lines in the text we are using the “Simple view” format, hiding the track changes. 

1. Line 41: Cost of diabetes per person: is this cost a one-time all cost? How is it appropriated?

Thank you for this comment. We have slightly reworded the text in lines 43-44 at the introduction section, as this estimate refers, indeed, to health expenditures, and not to costs.

In lines 43-45 the text is now: “In Spain, the mean cost for the healthcare system of treating a patient with diabetes, aged between 20 and 79 years, is estimated at $2,651.5 (USD) [2].”

Regarding how these expenditures are calculated: this is the result of an estimation provided by the latest international Diabetes Federation Atlas, at its Factsheet for Europe. It can be found at:

https://diabetesatlas.org/upload/resources/material/20191218_144548_eur_factsheet_en.pdf

2. Could the authors provide some brief background information about the Basque population? It is important to clarify what are the main characteristics and/or special characteristics of the population.

Thank you for this comment. We have included a paragraph that explains briefly the characteristics of the Basque Country population participating in the study, and why is the Basque Country a lead region in delivering care, and also chronic care. We have also included some arguments regarding this in the discussion section.

At the Introduction section,  in lines 53-59, we included: “The Basque Country provides public healthcare to over two million inhabitants in the region, and around 18% are aged 65 or over [13]. The Basque Country has a Strategy for Tackling the Challenge of Chronicity [14]. It contains policies and projects aimed at reinventing the health delivery model with the purpose of improving the quality of care for chronic patients and advancing towards a more sustainable, proactive, and integrated model. The strategy includes plans to address different chronic problems including secondary prevention and intensified follow-up of patients with diabetes.”

3. Consider providing a map identifying the location/geography pertinent to the studied population.

Thank you for this comment, our population are individuals living in the Basque Country, that responded to the Basque Health Survey in 2018. We believe it is not necessary to include much detail regarding the geographical location of this Spanish region, but we made an effort to clarify in the text who our population is, and their characteristics.

To clarify this, we have slightly rewritten the paragraph in section 2.1 The Basque Health Survey, in lines 88-89: “The Basque Health Survey (ESCAV) is conducted by the Basque Government every 5 years with people living in the Basque Country.”

4. The studied sample of 555 participants appears to cross from adolescent to adulthood. How do authors correct for this age crossover, especially when there is both underage and adult participants?

Thank you for this comment. As shown in Table 1 (descriptive analysis), there are no adolescents in our sample. Our respondents are all people who declared having diabetes in the Basque Health Survey, and there are no respondents under 25 years of age (note that the Basque Health Survey does include people 15+, but when we select the people with declared diabetes, there are no respondents aged 15-24).

All this translated in that our youngest individuals, when we select those who have declared at least a diagnosis of diabetes, are the ones in the group 25-44 yrs.

To make this clear we have modified the following text in lines 99-100 in section 2.2 Design and working sample:

“This created a working sample of N= 555 respondents, aged 25 or older, reporting diabetes, and who live in the Basque Country.”

Regarding the second part of this comment:

To correct for the elderly groups, we created an additional group for identifying those aged 90+. We realised there were only 14 individuals in this group, but we thought it was important to separate them, and keep them in a different group, as they might be a group of individuals with a significantly greater number of comorbidities. In addition, we included, in our regression analyses, interactions of age and other sociodemographic characteristics.

Details of regression with interactions were submitted with the article as supplementary material because including all the controls and interactions the table was very difficult to read. However, if you think we should incorporate these in the table, we would be happy to substitute the current tables in the main text by the full tables that include all control variables and interactions.

5. What were the inclusion and exclusion criteria for the selection of the 555 cases studied? (i.e.: How were the 555 selected?).

 Thank you for this comment and sorry for the confusion. Lines 96-103 in the Material and Methods section have been rewritten to make clear the selection criteria of our working sample: people who declared having a diabetes diagnosis in the Basque Health Survey.

The text is now: “Our dataset includes information from individuals who have declared having at least one type of chronic problem that has been diagnosed by a medical doctor. However, because our work focused on people with diabetes-related problems, our sample consisted only of those individuals who declared having at least a diabetes diagnosis. This created a working sample of N= 555 respondents, aged 25 or older, reporting diabetes, and who live in the Basque Country. Data on self-assessed experience of care was obtained from the Basque Health Survey 2018, which also included self-reported sociodemographic and economic data.”.

6. How was the number 555 decided? Was there a power calculation or another method that led to the selection of 555 out of the total number of datapoints available in the database?

Thank you for this comment and sorry for the confusion. Hopefully this has been clarified with our response to your comment #5. This number of respondents was not calculated. The survey is considered as representative of the Basque general population and in that sense, the sampling methodology used is considered of strong power for making any calculations.

We have not incorporated weights to make any calculations for this paper, so these are raw numbers. This means we are using the actual number of people that had, at the moment of the survey, declared having a diagnosis of diabetes. If we would like to conduct a different type of analysis, such as making extrapolations of this population to the Basque region of some aspect, we could use weights, as they are available, given the fact that the population surveyed is considered representative of the general population in the Basque country. This was not the purpose of this study, and therefore we have not included any weighting for our analyses in this paper.

7. What were confounding factors and how were they controlled for (age, sex, body composition, BMI, years diagnosed with diabetes, smoking status etc) ?

Thank you for this comment. We initially considered as confounders the following variables:  Age, gender, net monthly income, level of education, occupation, and interactions. However, for the regression analyses, we had to make some decisions about what to include or not include, all these based on results from descriptive analyses (this is explained in the text, Methods section, in lines 191-199).

Regarding other confounding factors that you suggest, unfortunately we do not have the information on respondents’ Body Mass Index, although it looks like an obvious confounding factor to add. We also do not have information regarding the number of years with the disease, nor regarding the smoking status of respondents.

The reason why we do not have this information on BMI is because we did not ask for it and therefore, we were not provided with it. We did not include the years with diagnosis of the patient (this was not a variable of the questionnaire), neither the smoking status. The moment of requesting data is usually complicated. We had to prioritise and provide a conservative list of variables to conduct our study. Unfortunately, we did not realise of the importance of some variables when we had to make our choice of priority variables to request.

8. Please include a short section on strengths and limitation (limitations are actually mentioned briefly in the Discussion section) of the study.

Thank you for this comment. We have included a short section on strengths and limitations of this study. This is now section 4 (Study strengths and limitations) in the Manuscript, and Discussion section is now section 5, starting in line 365.

The way it is organised now is: section 4 presents the strengths and limitations briefly, and section 5 discusses those strengths and limitations, providing possible explanations and implications for each of the strengths and limitations presented in section 4. 

The text of the new section 4. Study strengths and limitations, in lines 337-362, is now:

“4. Study strengths and limitations

Our study’s main strength is that it is the first study that analyses patients’ evaluation of  their experiences with healthcare services across several factors, using a sample with prevalence rates of chronic conditions that is representative of the wider population. Despite the fact that chronic problems were not used to ensure representativeness of the population in the ESCAV 2018 survey[1], the prevalence rates of our participants who declared a diagnosis by a doctor of a chronic problem are almost identical to prevalence rates of patients who have a clinical diagnosis of chronic problems in the Basque Country. Therefore, we can assume that their experiences should be representative of the wider population of diabetes sufferers in the Basque Country.

The study does suffer from some limitations. First, reported chronic conditions are self-declared, although our evidence from diabetes suggests that these types of self-declared reports, where the question relies on a declaration of a diagnosis confirmed by a doctor, are reliable. Second, we count conditions to control for multimorbidity, but there is no information regarding the severity or the progression of the conditions. Third, we analyzed IEXPAC factors instead of conducting the analyses for each of the IEXPAC items, and therefore we could be missing important information if there are differences between the distributions of responses in each factor. However, our descriptive analyses showed almost identical results to previous literature, as explained below in the discussion. Fourth, other potential variables, such as the patient ethnicity or the health area responsible for providing care, which would be desirable confounding factors to include, were not available. Finally, the IEXPAC questionnaire, included in the ESCAV survey, asks questions regarding experience with healthcare services, but we cannot derive from responses if these are public or private services. Possible explanations and implications for these limitations are presented in the Discussion section, along with suggestions for further research.”

9. While the repository from which the authors drew cases is rather large the 555 sample is not necessarily deemed “corresponding/characteristic/representative” if the entire Basque population. What is the Basque population?

Thank you for this comment. The sample of people declaring diabetes problems (N = 555) at the Basque Health Survey (ESCAV 2018) is, indeed, not representative of the population with diabetes in the Basque country. The total population surveyed (N=8,036) is, though, representative of the Basque country population (Pop in 2018 = over 2 million inhabitants).

10. Please provide a section with brief description of the study which repository was used to select the cases studied in addition to the reference (website).

More info on the Basque Health Survey is available here: https://en.eustat.eus/document/encsalud_i.html. The dataset is available on reasonable request (for research purposes) to the Basque Government. The specific dataset of this study is not public but the authors have got a special permission to share it for this type of purpose. We have included all the answers to the Survey of patients with report diabetes based on a medical doctor diagnosis.

11. Line 187: please explain/correct “Error! Reference source not found”

 Thank you for this comment. We are sorry but we do not see this error in the manuscript.

12. The discussion section is rather short and most importantly does not discuss the findings in relation to other similar studies or other populations.

Thank you for this comment, we have expanded the discussion section. Specifically, we have incorporated some explanations on how our results confirm/contradict what other studies have found.

13. Could the results be interpreted differently? Is there a behavioral or social theory that could explain the results?

 Thank you for this comment. We have added a comment in the discussion regarding the possibility of a behavioural explanation regarding the impact of chronic diabetes problems on healthcare experience.

In lines 408-417 at the Discussion section, we added:

“This could have a behavioural explanation.

Our population of respondents may not have that much choice and even if they do have choice, they usually do not exert it. Chronically ill patients often prefer, for example, the nearest hospital, simply for convenience [30], even if it is not the best hospital in their hospitals’ choice set. They may not even be aware that there could be better care available if they travelled further. Additionally, with a chronic condition such as diabetes, the consequences of not receiving any treatment are worse than the consequences of receiving poor care, and therefore diabetes patients may be more accepting of “customer service” shortcomings than patients with other problems where bad “customer service” may play a more important role in the patients’ assessment of their care.”.

14. The healthcare system through which care was received was it uniform? Were there patients receiving care from private versus public clinics? Was the accessibility financial, geographical, physical, cultural, race/ethnicity factors etc considered? All these parameters can impact the level of stress and anxiety and as a result the rating of the healthcare experience.

Thank you for this comment. Experience is regarding the utilisation of healthcare services, but we do not know if these are public or private. Although it might be a good hypothesis to test if people with private insurance report worse experience with the public setting, this is something that cannot be tested with this dataset. We have added this into the discussion section as it seems a very pertinent discussion point.

In lines 358-360 in section 4. Study strengths and limitations, we added:

“Finally, the IEXPAC questionnaire, included in the ESCAV survey, asks questions regarding experience with healthcare services, but we cannot derive from responses if these are public or private services.”.

15. Please include a formulated and clearly articulated hypothesis (Ho).

Thank you for this comment. We have formulated the hypothesis of this study in the manuscript. This is now a new subsection of the Material and methods section: 2.5 Hypotheses. The new text is in lines 160-171:

“In this study, we test the following hypotheses:

  • If, in the Basque Country, there are inequalities in reported experiences of healthcare amongst people with chronic diabetes problems, according to individual’s sociodemographic characteristics (such as gender, age, education or occupation).
  • If, in the Basque Country, and among people with diabetes problems, there are some chronic comorbidities that can be associated with worse reported experiences of healthcare than others.
  • If, in the Basque Country, and among people with diabetes problems, those with multiple other chronic comorbidities report worse healthcare experiences than those with a lower number of chronic comorbidities.”.

16. The manuscript has a relatively limited list of references. More studies could be considered and cited. Some works to consider:

Sikalidis AK, Öztağ M Optimized snacking is positively associated with socioeconomic status and better Type 2 Diabetes Mellitus management in Turkish patients. Gazz Med Ital - Arch Sci Med. 2020; 179(7-8):459-67. doi:10.23736/S0393-3660.19.04159-9.

Volaco A, Cavalcanti AM, Filho RP, Précoma DB. Socioeconomic Status: The Missing Link Between Obesity and Diabetes Mellitus? Curr Diabetes Rev. 2018;14(4):321-326. doi: 10.2174/1573399813666170621123227.

Yin T, Yin DL, Xiao F, Xin QQ, Li RL, Zheng XG, Yang HM, Wang LH, Ding XY, Chen BW. Socioeconomic status moderates the association between patient satisfaction with community health service and self-management behaviors in patients with type 2 diabetes: A cross-sectional survey in China. Medicine (Baltimore). 2019 May;98(22):e15849. doi: 10.1097/MD.0000000000015849.

Thank you for this last comment. We have incorporated some of these references (Volaco et al., 2018) as they were, indeed, relevant for the context setting of this paper. We have not considered the paper by Sikalidis et al. as, in the former, the research question is different to the question assessed in our study. We included Yin et al. paper, but note that this paper is focused on patients’ satisfaction, which is a measure related with experience, but is not the same as experience (in our paper we measure experience). We also have added some new references after responding to some of your comments in the background and discussion sections.

Reviewer 2 Report

This study analyzes the experience with the health care system by a representative sample of patients with self-reported diabetes from the Basque Country. The results show that patients in general value aspects on new ways of interaction through internet or with peers (NEW) worse than aspects on productive interactions with professionals (INTER) or on self-management ability (SELF). Having comorbidities significantly decreases the valuation of experience in all three categories, and older patients valuate experience better that younger patients. The number of comorbidities appears to be a more important determinant than any specific comorbidities. The authors conclude that implementing new technologies and internet resources is warranted to increase the quality of diabetes care. The paper is well written and the statistical methods to analyze the data appear state-of-the-art. There are some minor issues.

  1. Abstract: Why did the authors choose not to include the worse outcome of the NEW items compared to the INTER or SELF items as results or in the conclusion sentence?  Isn’t the statement in lines 339/340 the most important conclusion of this study?
  2. Introduction lines 34-36: how do the two figures on incidence (11.6 and 3.7) relate to each other? Is the 9.12 % referring to incidence or prevalence?
  3. Methods line 85: please elaborate on how this sample of 555 respondents was selected from the 2018 survey.

Line 119-126: by far the most respondents have probably retired, how is occupation categorized for them?

  1. Results: line 181: Did 70+% of the respondents fill in the 4 items of this category, or did they score the 4 items with “always” or “mostly”?

Table I and Table 2: Please explain what is meant with the column "mean", and what with "SD" in this column? The scores for the Factor 2 NEW do not seem to be normally distributed, is mean (SD) appropriate? Part of the text underneath table 2 refers to Table 1.

Why are the variables in table 2 not ordered according to the number of observations (diabetes on top?)?

Line 217/218: the 10.27% of respondents does not seem to correspond with the data in Table 2.

Table 3: why is sample size given with three digits after the decimal point?

Line 245 and 290: please elaborate what the number -2.073 and +11.60, respectively, refer to.

Table 4: it would be more informative if the categories are listed in the same order as in Table 2.

Lines 265-268: please consider to stress here that constipation and other mental conditions are only reported by a very small minority of respondents.

  1. Discussion: lines 343-344: similar to above

Author Response

Responses to Reviewer 2

This study analyzes the experience with the health care system by a representative sample of patients with self-reported diabetes from the Basque Country. The results show that patients in general value aspects on new ways of interaction through internet or with peers (NEW) worse than aspects on productive interactions with professionals (INTER) or on self-management ability (SELF). Having comorbidities significantly decreases the valuation of experience in all three categories, and older patients valuate experience better that younger patients. The number of comorbidities appears to be a more important determinant than any specific comorbidities. The authors conclude that implementing new technologies and internet resources is warranted to increase the quality of diabetes care. The paper is well written and the statistical methods to analyze the data appear state-of-the-art. There are some minor issues.

Thank you for the time you devoted to read, comment and improve our paper. We appreciate your positive critique of the paper. We believe that your comments were relevant and essential to improve the quality of the paper. We now respond, one by one, to each of your comments. Please, note that when referring to specific lines in the text we are using the “Simple view” format, hiding the track changes. In addition, please note that the resubmitted version of the manuscript includes the English revision suggested. 

1. Abstract: Why did the authors choose not to include the worse outcome of the NEW items compared to the INTER or SELF items as results or in the conclusion sentence?  Isn’t the statement in lines 339/340 the most important conclusion of this study?

Thank you for this comment. We agree with you on that this is an important result, and we have included this result in the conclusions.

However, because we already mention this in lines 249-251 in the Results section, in order to avoid redundancy, we removed the last sentence in line 446 in the discussion section.

In lines 469-474 we included:

“Our population, people living in the Basque Country with at least a declared chronic diabetes problem, reports, overall, good levels of experience with most of the analysed aspects of the healthcare received. However, we also observe lower experience ratings with the new relational model factors, which suggests that there is still work to do on improving the new technologies and internet resources offered to the population we have studied.”

 2. Introduction lines 34-36: how do the two figures on incidence (11.6 and 3.7) relate to each other? Is the 9.12 % referring to incidence or prevalence?

Thank you for this comment. The 9.12% is a prevalence rate, indeed. We have reviewed this paragraph in the introduction.

In lines 34-40 the corrected text is now:

“The Di@bet.es nationwide population-based cohort study in Spain showed an incidence of Type 2 diabetes of 11.6 cases/1000 person-years, and an incidence of known diabetes of 3.7 cases/1000 person-years in Spain [3]. The prevalence of Type 2 diabetes mellitus was 9.12% amongst all citizens aged ≥35 in the Basque Country in 2011 [4], and 10.6% according to a study published in 2017 [5]. This Basque Country prevalence rate (10.6% in 2017) is lower than the prevalence rate for Spain, which was estimated as 13.8% according to results from a recent systematic literature review [6].”

3. Methods line 85: please elaborate on how this sample of 555 respondents was selected from the 2018 survey.

Thank you for this comment. Lines 95-102 in the Material and Methods section (in section 2.2 Design and working sample) have been rewritten to make clear the selection criteria of our working sample: people who declared having a diabetes diagnosis in the Basque Health Survey.

“We conducted a cross-sectional study. Our dataset includes information from individuals who have declared having at least one type of chronic problem that has been diagnosed by a medical doctor. However, because our work focused on people with diabetes-related problems, our sample consisted only of those individuals who declared having at least a diabetes diagnosis. This created a working sample of N= 555 respondents, aged 25 or older, reporting diabetes, and who live in the Basque Country. Data on self-assessed experience of care was obtained from the Basque Health Survey 2018, which also included self-reported sociodemographic and economic data.”

4. Line 119-126: by far the most respondents have probably retired, how is occupation categorized for them?

Thank you for this comment. Our respondents were asked about their current or past employment. There is not a category, within the type of occupation, for the retired. There are not missing values either for occupation, which means our respondents distribute across the categories considered in the paper, which are exactly the same categories available in the questionnaire. Thus, if an individual is retired, the response should be interpreted as the type of occupation that this individual had most of his/her working life.

In order to clarify this, we have added an explanation in lines 143-145 of the current version of the manuscript:

“Note that occupation might be current (for people of working age) or past (for retired individuals), and the same categories apply for both types of respondents.”

5. Results: line 181: Did 70+% of the respondents fill in the 4 items of this category, or did they score the 4 items with “always” or “mostly”?

Thank you for this comment. It is indeed the % of people responding “always” or “mostly” to these items. We have reviewed the text, now in lines 211-214:

“More than 70% of the patients responded “always” or “mostly” to the items that related to productive interactions (Factor 1: items 1, 2, 5 and 9), and self-management abilities (Factor 2: items 4, 6, 8 and 10), except for Factor 2, item 10, where the percentage is below 40%.”.

6. Table I and Table 2: Please explain what is meant with the column "mean", and what with "SD" in this column? The scores for the Factor 2 NEW do not seem to be normally distributed, is mean (SD) appropriate? Part of the text underneath table 2 refers to Table 1.

Thank you for this comment. We have included an explanation of what the mean for each variable represents in Tables 1 and 2.

In lines 221-226: “The mean for each variable is provided, together with its standard deviation. Given that we have converted all variables into categorical variables, the mean value of a certain category can be also interpreted as the proportion of respondents in that specific category. For example, the mean for women is 0.463, and the mean for men is 0.537. The sum of both means equals 1, which means that 0.463 is the proportion of women (or 46.3%) with respect to the total number of respondents in our study population.”.

Regarding the normality issue for factor 2 (NEW), we see your concern. Although responses to this factor show very poor experience, deviations are also small. We, thus, do not see why using another measure would improve our analysis. In addition, our results need to be comparable, and comparability could be lost if we would incorporate a different measure for this specific factor just because its distribution is not “perfectly” normal. For what we seen, our problem is not an asymmetry problem, but our distribution is just positioned to the left, as respondents give low punctuations to this factor. This is not new, and has been observed previously in the literature, as we mention in the discussion. However, if you have a specific suggestion on another way to present this factor, we would be happy to test if it makes sense to try it and incorporate it if you believe it is definitely something that will add value to the paper. 

Regarding your last point, we have reviewed the text underneath Table 2 and removed the text that, indeed, was specific of Table 1.

7. Why are the variables in table 2 not ordered according to the number of observations (diabetes on top?)?

Thank you for this comment. We have moved Diabetes to the top of the table. The rest of the table was already ordered according to the number of respondents with each chronic problem.

8. Line 217/218: the 10.27% of respondents does not seem to correspond with the data in Table 2.

Thank you for this comment. It is 11.41%. We have corrected this in the text (line 247-248 in the current reviewed version): “…these two conditions affect 11.41% of the respondents…”

9. Table 3: why is sample size given with three digits after the decimal point?

Thank you for this comment. This typo is now corrected.

10. Line 245 and 290: please elaborate what the number -2.073 and +11.60, respectively, refer to.

Thank you for this comment. We have included an explanation right after the mentioned numbers. These refer to the interaction effects, which are not visible in the tables in the main text, but are shown in the supplementary material instead. We decided to put the full regression result tables as supplementary materials, as they are very long for fitting in a single page if we show all interaction effects included. However, if you think we should reconsider including the full table in the main text of the manuscript, please, let us know.

11. Table 4: it would be more informative if the categories are listed in the same order as in Table 2.

Thank you for this comment. We have ordered Table 4 using the same ordering criteria than in Table 2. We accepted the track change in this case, as this can be considered a change of format, and we left a comment indicating the ordering change.

12. Lines 265-268: please consider to stress here that constipation and other mental conditions are only reported by a very small minority of respondents.

Thank you for this comment. We have added a comment in the text. In lines 303-306: “These are the only two conditions which have significant effects at the 95% level. One should note, however, that constipation and other mental conditions are only reported by a very small proportion of respondents, and therefore, their impact on the population experience levels will be small.

13. Discussion: lines 343-344: similar to above

Thank you for this comment. Responding to this comment, in lines 453-454 we have added the following sentence: “These, however, were reported only by a small proportion of respondents, so their impact on the experience, at the population level, is small.".

Round 2

Reviewer 1 Report

The authors have addressed the reviewer's points in a satisfactory manner. The manuscript is significantly improved and the authors should be commended on their efforts. Minor spellcheck and overall proofreading is recommended.